# Salt Stress Inhibits Photosynthesis and Destroys Chloroplast Structure by Downregulating Chloroplast Development–Related Genes in *Robinia pseudoacacia* Seedlings

**DOI:** 10.3390/plants12061283

**Published:** 2023-03-11

**Authors:** Chaoxia Lu, Lingyu Li, Xiuling Liu, Min Chen, Shubo Wan, Guowei Li

**Affiliations:** 1Shandong Provincial Key Laboratory of Crop Genetic Improvement, Ecology and Physiology, Shandong Academy of Agricultural Sciences, Jinan 250100, China; 2Shandong Provincial Key Laboratory of Plant Stress Research, College of Life Science, Shandong Normal University, Jinan 250014, China; 3Dezhou Graduate School, North University of China, Kangbo Road, Dezhou 253034, China

**Keywords:** *Robinia pseudoacacia* seedlings, photosynthesis, chloroplast, salt stress

## Abstract

Soil salinization is an important factor limiting food security and ecological stability. As a commonly used greening tree species, *Robinia pseudoacacia* often suffers from salt stress that can manifest as leaf yellowing, decreased photosynthesis, disintegrated chloroplasts, growth stagnation, and even death. To elucidate how salt stress decreases photosynthesis and damages photosynthetic structures, we treated *R. pseudoacacia* seedlings with different concentrations of NaCl (0, 50, 100, 150, and 200 mM) for 2 weeks and then measured their biomass, ion content, organic soluble substance content, reactive oxygen species (ROS) content, antioxidant enzyme activity, photosynthetic parameters, chloroplast ultrastructure, and chloroplast development-related gene expression. NaCl treatment significantly decreased biomass and photosynthetic parameters, but increased ion content, organic soluble substances, and ROS content. High NaCl concentrations (100–200 mM) also led to distorted chloroplasts, scattered and deformed grana lamellae, disintegrated thylakoid structures, irregularly swollen starch granules, and larger, more numerous lipid spheres. Compared to control (0 mM NaCl), the 50 mM NaCl treatment significantly increased antioxidant enzyme activity while upregulating the expression of the ion transport-related genes Na^+^/H^+^ exchanger 1(*NHX 1*) and salt overly sensitive 1 (*SOS 1*) and the chloroplast development-related genes *psaA*, *psbA*, *psaB*, *psbD*, *psaC*, *psbC*, *ndhH*, *ndhE*, *rps7*, and *ropA*. Additionally, high concentrations of NaCl (100–200 mM) decreased antioxidant enzyme activity and downregulated the expression of ion transport- and chloroplast development-related genes. These results showed that although *R. pseudoacacia* can tolerate low concentrations of NaCl, high concentrations (100–200 mM) can damage chloroplast structure and disturb metabolic processes by downregulating gene expression.

## 1. Introduction

Soil salinization is already a global problem that climate change, inappropriate irrigation methods, and natural disasters, such as hurricanes and tsunamis, threaten to exacerbate [1,2]. It is estimated that about 70% of the world’s agricultural drylands (5.2 billion hectares) are affected by erosion, soil degradation, and salinization [3]. Saline environments account for nearly 10% of the world’s total land area (950 Mha), and 50% of irrigated land (230 Mha), which seriously impedes agricultural development [4,5]. It is therefore urgent that we develop and utilize saline areas to solve the problem of land shortages and ensure food security [6,7].

High concentrations of salt ions in soil cause osmotic stress, ion toxicity, and nutrient imbalance in plants, which can seriously disrupt normal growth [8,9]. Salt ions in plants can also damage organelle structure; affect cellular processes including photosynthesis; alter mRNA and protein synthesis; and disrupt energy metabolism, amino acid biosynthesis, and lipid metabolism [10,11]. In addition, salt stress causes reactive oxygen species (ROS) accumulation, causing oxidative stress in plants. Some halophytes and salt-tolerant plants can eliminate ROS by increasing the content and activity of antioxidant enzymes under salt stress, thus reducing damage to the plant. However, most crop plants cannot remove ROS effectively, leading to metabolic disorders and inhibited plant growth [12,13,14].

Photosynthesis, as the source of energy for plant growth and development, is the most basic and critical physiological process in green plants [15,16,17]. Salt stress seriously affects photosynthesis in plants [18,19]. Large amounts of Na^+^ and Cl^−^ in the leaves leads to water loss from guard cells and therefore to changes in guard cell and stomata morphology, activity, and/or density numbers, which can reduce or prevent CO_2_ entry and limit normal photosynthesis [20,21,22]. Moreover, salt stress leads to ion toxicity and oxidative stress, particularly in chloroplasts, the main sites of ROS production. Salt stress affects the electron transport chain (ETC) from the photosynthetic complex photosystem II (PSII) to PSI, resulting in the copious production of ROS that causes oxidative damage to nucleic acids, proteins, lipids, membranes, and some photosynthetic enzymes, reducing CO_2_ oxidation and crop yield [23,24]. Salt stress can also damage the structural integrity of chloroplasts, which is essential to the photosynthetic light reactions and carbon assimilation processes [25]. Under salt stress, salt-tolerant plants can maintain the normal morphology of photosynthetic structures to sustain their functions [26]. However, when salt stress intensity exceeds the tolerance capacity of the plants, various organelles are damaged, accompanied by changed ultrastructure. Studies showed that under salt stress, the chloroplast membrane system was damaged, with the shape of the chloroplasts changing from ovoid to spherical, the inner lamellae becoming slightly swollen, the number of basal lamellae decreasing, and the number of starch grains, lipid droplets, and osmiophilic globules in the chloroplasts increasing [27]. Many genes involved in the development and maintenance of the normal chloroplast structure that is integral to photosynthesis are known, and salt stress has been shown to affect the expression of such genes, leading to aberrant chloroplast structure. Huang et al. (2019) found that the expression of *Ndhf* genes related to light response, *Rbcl* genes related to dark response, and *Matk* genes related to chloroplastic intron splicing were downregulated under 150 mM NaCl in *Eucalyptus robusta* chloroplasts, which led to decreased photosynthesis and deformed chloroplasts [28,29,30].

*Robinia pseudoacacia* L. also known as black locust or false acacia, is native to the United States and belongs to the Faboideae subfamily of the Fabaceae. *R. pseudoacacia* is often used as an afforestation species in saline areas because it reproduces easily and grows rapidly, while also having a high tolerance for salinity, drought, soot, and dust. Therefore, it plays an important role in improving saline environments, preventing soil erosion, and regulating hydrology [31,32]. However, Lu et al. (2021) found the severe degradation of black locust forests in the Yellow River Delta, manifested by yellowing leaves and dead tree tips [33]. This suggests that salt stress can affect photosynthesis in black locusts, but there is no report on how salt stress affects chloroplast structure or how chloroplasts adapt to changes during salt stress. In this study, we treated *R. pseudoacacia* seedlings with different concentrations of NaCl to elucidate these mechanisms by measuring photosynthetic indicators, chloroplast structure, and changes in gene expression.

## 2. Materials and Methods

### 2.1. Plant Material and Seed Germination

The *R. pseudoacacia* seeds were Luci 103 (provided by Daqingshan forest farm in Feixian County, Shandong Province). Seeds were sterilized in 75% ethanol for 15 min, then in NaClO (3%) solution for 15 min, and washed with sterile water 3 times. Sterilized, washed seeds were placed in a culture bottle with sterile water, which was put into an 85 °C water bath for 15 min followed by a 45 °C water bath for 12 h.

Seeds were evenly sown in nutrient soil in a greenhouse (25 ± 3 °C/22 ± 3 °C, day/night) at a light intensity of 600 mol/m^2^/s (with a 15 h photoperiod) and a relative humidity of 60/80% (day/night). The seedlings were irrigated with water, and germinated for 3–5 days, and NaCl treatment was started when the seedlings had grown to 13–15 cm in length. Five NaCl concentrations of 0, 50, 100, 150, and 200 mM were used, with five replicates per treatment. Plants were watered regularly with 200 mL per pot at 8:30 am daily. To avoid salt shock, salt treatments were increased gradually (increasing by 50 mM per day) until the final desired concentration was reached (the fourth day). Measurements were taken after 2 weeks at that concentration.

### 2.2. Measurement of Plant Height, Root Length, Dry Weight, Fresh Weight and Water Content

Plant height and root length of *R. pseudoacacia* seedlings were measured to the nearest millimeter. Fresh weights of the shoots and the roots (FW1 and FW2, respectively) were taken, and then the samples were heated to 105 °C for 20 min. Samples were then kept in an 80 °C oven until weights remained constant (about two days), and then their dry weights (DW1, DW2) were measured.
Water content of shoot = (FW1 − DW1)/FW1 × 100%
Water content of root = (FW2 − DW2)/FW2 × 100%

### 2.3. Extraction and Quantification of Ion Contents

Leaf samples (0.3 g) were weighed, boiled in water for 3.5 h, and then filtered to a constant volume of 25 mL. Na^+^ and K^+^ contents were measured using a flame spectrophotometer (Model 2655-00 Digital Flame Analyzer; Cole-Parmer Instrument Co. Vernon Hills, IL, USA). Cl^−^ content was measured using an ICS-1100 ion chromatography system (Dionex Corporation, Sunnyvale, CA, USA).

### 2.4. Determination of Soluble Organic Matter

A 0.3 g subsample of fresh leaves was ground together with 2.4 mL of prepared phosphate-buffered saline (PBS) solution (pH 7.8) and then centrifuged at 10,000× *g* for 10 min at 4 °C.

The concentration of total soluble sugars was measured using the anthracene-sulfur colorimetric method [26]. Soluble protein content was determined using Coomassie Brilliant Blue G250 and BSA [34]. Proline content was determined using ninhydrin [35]. Fresh leaves from the same plants were also ground with 5 mL dH_2_O, heated in a water bath at 85 °C for 30 min, allowed to cool, and filtered, and the solution volumes were standardized to 0.1 L. Six drops of phenolphthalein solution (1%) were added to 0.05 L of the extract and titrated with NaOH solution (10 mM) until it turned pink. The volume of NaOH required was recorded and used to calculate the content of organic acids.

### 2.5. Determination of MDA, O_2_^−^, H_2_O_2_

The fully expanded leaves of the third branches of *R. pseudoacacia* seedlings subjected to different NaCl treatments were used to measure H_2_O_2_, O_2_^–^, and malondialdehyde (MDA). Detailed experimental procedures were described below.

Leaves of 0.3 g FW were cut into pieces and ground with 2 mL 1% trichloroacetic acid solution and placed in a test tube containing 0.5% thiobarbituric acid solution. The test tube was boiled for 10 min and then the solution was filtered. OD values of the filtrate were measured at 532 nm and 600 nm to determine malondialdehyde content [36].

Leaves of 0.5 g FW were cut into pieces and ground with 5 mL 50 mM PBS (pH 7.8) at 4 °C. The solution was filtered, and the filtrate was placed in a test tube containing 5 mL reaction solution (17 mM aminobenzenesulfonic acid: 7 mM α-naphthylamine, 1:2 (*v*/*v*)). This was allowed to stand for 30 min after mixing and then the OD value at 530 nm was measured to determine O_2_^−^ content [37,38].

Leaves of 0.5 g FW were cut into pieces and ground with 5 mL acetone at 4 °C. The solution was centrifugated at 5000× *g* for 8 min. A 2 mL supernatant was placed in a test tube containing 0.1 mL 20% TiCl_4_ and 0.4 mL strong ammonia solution and then mixed and centrifugated at 8000× *g* for 7 min. After being washed 3–5 times with acetone, the precipitate was dissolved with 5 mL 2 M H_2_SO_4_ and the OD value 415 nm was measured to determine the H_2_O_2_ content [39,40].

### 2.6. Determination of Electrical Conductance

The relative electric conductivity (REC) was operated according to Guo et al. (2015) [41]. Leaves were cleaned and punched into leaf discs. The leaf discs were soaked in dH_2_O for 2 h, which was called solution 1. The initial conductance of solution 1 was measured with a conductivity measuring instrument (DDG-5205A, Leici, Shanghai, China) and the conductance value is represented by L1. Solution 1 then was placed in a boiling water bath for 15–20 min and was then called solution 2. Solution 2 was cooled to room temperature and the leaf discs were removed. The conductance of solution 2 was represented by L2. The REC was calculated using the following formula:REC = (L1/L2) × 100%

### 2.7. Antioxidant Enzyme Activity Assays

Fresh leaves (0.3 g) were ground in PBS solution (2.4 mL) and then centrifuged at 10,000 rpm for 10 min at 4 °C; the supernatant was then used as the enzyme solution for testing.

The superoxide dismutase (SOD) activity of the enzyme solution was measured by the nitro blue tetrazolium (NBT) photochemical reduction method [42,43], and was detected using an Total Superoxide Dismutase Assay Kit (S0109, Biyuntian, Shanghai, China) according to the manufacturer’s protocols. Peroxidase (POD) activity was measured using guaiacol [44]. With guaiacol as a substrate, the changes of absorbance at 470 nm were measured after an enzymatic reaction, the amount of enzyme required for each 1 min absorbance change of 0.01 is one unit of activity. Catalase (CAT) activity was measured in 0.2 mL enzyme solution by adding 6 mL of reaction solution, which consisted of 200 mL PBS with 0.3092 mL of 30% H_2_O_2_ [43]. Ascorbate peroxidase (APX) activity was determined using 0.1 mL enzyme solution by adding 1.7 mL of PBS containing EDTA-Na_2_ (0.1 mM), followed by 5 mM ascorbate (ASA) (0.1 mL), adding 20 mM H_2_O_2_ (0.1 mL), and immediately (within 40 s) measuring the light absorption at 290 nm wavelengths [45].

### 2.8. Measurement of Gas Exchange Parameters

After exposure to different NaCl concentrations for 15 days, photosynthetic parameters were measured from 14:00 h to 17:00 h using a Li-6000 portable photosynthesis system (Li-Cor, Lincoln, NE, USA). Five replicates were used for each treatment.

### 2.9. Preparation of Temporary Mount on the Leaf Surface of R. pseudoacacia

The leaves of *R. pseudoacacia* were put into Carnot fixative fluid (3:1 (*v*/*v*) ethanol:glacial acetic acid) for 1 h, and the temporary mount was prepared by adding Hoyers solution (40 mL lactic acid with chloral hydrate until saturation) for more than 30 min, which was observed under a differential interference (DIC) microscope (ECLIPSE80i Differential Interference Fluorescence Microscope, Nikon, Japan).

### 2.10. Ultrastructure of Chloroplasts in Seedlings of R. pseudoacacia

A 1 cm^2^ sample of leaf tissue was placed in 2.5% glutaraldehyde (pH = 6.8) fixative fluid. The samples were immersed in the fixative by evacuating the air and left for 2 h at 4 °C. The fixed samples were removed and rinsed with 0.1 mol/L phosphoric acid buffer (pH = 6.8) at 2 h intervals (4 °C). This was followed by sequential dehydration with 50%, 70%, 95%, and 100% ethanol. Dehydrated samples were transferred to 100% acetone for displacement, then epoxy-impregnated, encapsulated, and polymerized. Sections were made with an ultrathin sectioning machine (Leica EM UC7, Weztlar, Germany). The sections were then double stained with uranyl acetate dioxide; lemon leaf was used and examined under a transmission electron microscope (Hitachi HT7800, Tokyo, Japan).

### 2.11. qRT-PCR Analysis

After NaCl treatment, the leaves of *R. pseudoacacia* seedlings were collected at varying time points (0 h, 24 h, 48 h, 72 h) for measuring gene expression and then immediately frozen in liquid nitrogen and stored at −80 °C. Samples for each time point were taken from the same plant and five biological replicates at a time. The chloroplast development-related gene sequences (*RppsbA*, *RppsbD*, *RppsaA*, *RppsaB*, *RpndhE*, *RpndhH*, *RppsaC*, *RppsbC*, *Rprps7*, and *RpropA*) and ion transporter genes (*NHX1*, *SOS1*) of *R. pseudoacacia* were obtained from the NCBI website, and primers were designed using primer 5.0 (Appendix A). Total RNA was extracted using the Rapid Universal Plant RNA Extraction Kit 3.0 (Hua Yueyang, Beijing, China). cDNA synthesis was performed using the *Evo M-MLV* RT Mix Kit with gDNA Clean for qPCR (Code No. AG11728). Real-time quantitative reverse-transcription PCR (qRT-PCR) was performed using the SYBR Green Premix *Pro Taq* HS qPCR Kit (Accurate Biology, Changsha, China). The relative expression of each gene was calculated using the 2^−ΔΔCt^ method to calculate the relative expression of each gene [46]. The housekeeping gene *TUBULIN* was used as a control.

### 2.12. Statistical Analysis

Statistical analyses were made using SPSS software (version 19.0; IBM, Armonk, New York, NY, USA). Significance (*p* < 0.05) was determined using a Duncan’s multiple range test [47,48].

## 3. Results

### 3.1. Effect of Nacl Stress on the Growth, Ions Content and Organic Soluble Substances Contents of R. pseudoacacia Seedlings

NaCl stress significantly inhibited the growth of *R. pseudoacacia* seedlings. Although we did not detect significant differences from the 0 mM NaCl control at 50 mM NaCl, at higher NaCl concentrations (100, 150, and 200 mM), plant height, root length, shoot fresh weight, shoot dry weight, root fresh weight, and root dry weight all significantly decreased with increasing NaCl (Figure 1).

Compared with those in the control, the Na^+^ content, Cl^−^ content, and Na^+^/K^+^ ratio significantly rose with increasing NaCl concentrations, while the K^+^ content fell significantly (Figure 2A–D).

Compared with the control values, soluble protein, soluble sugar, and proline contents significantly increased with increasing NaCl concentration (Figure 3). Organic acid content showed no significant difference at 50 mM NaCl compared to the control, but rose at higher salt concentrations (100 mM, 150 mM, 200 mM). The increase in proline content was significantly greater than the increases for other osmotic regulators. At 50 mM, 100 mM, 150 mM, and 200 mM treatments, the proline contents were 1.31 times, 2.17 times, 3.36 times, and 3.68 times those of the control, respectively.

### 3.2. Effect of NaCl Stress on MDA, H_2_O_2_, and O^−^_2_ Contents and Electrical Conductance and Antioxidant Enzyme Activities of R. pseudoacacia Seedlings

Compared with those in the control, the MDA, H_2_O_2_, and O_2_^−^ contents and the electrical conductivity significantly increased along with increasing salt concentrations (Figure 4), indicating that a large amount of ROS was produced and suggesting the plasma membrane may have suffered damage.

With increasing salt concentrations, the activities of the antioxidant enzymes CAT, APX, SOD, and POD first increased and then decreased compared with the control (Figure 5). All four enzymes showed their highest activities at 50 mM NaCl. The activities of CAT and APX were also elevated at 100 mM NaCl, but reduced under higher-salt conditions (150 mM NaCl and 200 mM NaCl), while the activity of SOD increased at 100 and 150 mM NaCl and then decreased at 200 mM NaCl. Notably, the activity of POD showed the sharpest drop-off under high-salt conditions, being decreased at 100, 150, and 200 mM NaCl.

### 3.3. Effect of Nacl Stress on Photosynthesis, Chlorophyll Content and Leaf Grease Content and Chloroplast Structure of R. pseudoacacia Seedlings

Compared with those in the control, the net photosynthetic rate (Pn), transpiration rate (Tr), and stomatal conductance (Gs) significantly decreased with increasing NaCl concentrations. When treated with 200 mM NaCl, Tr, Pn, and Gs were 43.2%, 40.1%, and 41.1% of those of the control group, respectively. It can be seen that the inhibition effect of salt stress on photosynthetic parameters is obvious. However, intercellular CO_2_ concentration (Ci) increased with increasing NaCl concentration. When treated with 200 mM NaCl, Ci was 1.61 times higher than that of the control group. This indicates that the ability of gas exchange between plants and the outside world is inhibited under salt stress. Ci significantly increased, suggesting that NaCl stress inhibited the reduction of CO_2_ (Figure 6).

With increasing salt concentrations, total chlorophyll, chlorophyll a, and chlorophyll b contents significantly decreased compared to the control (Figure 7). At 200 mM NaCl, the total chlorophyll, chlorophyll a, and chlorophyll b contents were 55.2%, 47.1%, and 48.7% of the control group abundances, respectively.

With increasing salt concentrations, both the grease content and the degree of chloroplast structural damage increased compared to the control (Figure 8A,B). After 0 mM, 100 mM, and 200 mM NaCl treatment, there was grease content in the leaves of *R. pseudoacacia* seedlings. Under the control condition, the leaves of black locust seedlings had a small amount of grease content. However, with the increase in NaCl concentration, the grease content increased significantly. Further observation showed that the chloroplast structure changed with the increase in salt concentration. At 100 mM NaCl, the chloroplasts began to distort, the grana lamellae scattered and deformed, the structure of the thylakoids began to disintegrate, the starch granules started to become larger, swollen, and irregular, and the number and volume of lipid spheres increased. At 200 mM NaCl, the chloroplast structure further swelled and deformed, the thylakoids continued to expand and became disorderly, distorted, and deformed, filamentous lamellae appeared, starch granules continued to become larger and more irregular, and the number and volume of lipid spheres increased further.

### 3.4. Effect of NaCl Stress on the Expression of Key Genes in Chloroplast Development and Ions Transporter of R. pseudoacacia Seedlings

The relative expression of key genes (*RppsbA*, *RppsbD*, *RppsaA*, *RppsaB*, *RpndhE*, *RpndhH*, *RppsaC*, *RppsbC*, *Rprps7*, and *RpropA*) during the chloroplast development of *R. pseudoacacia* seedlings were significantly upregulated under 50 mM NaCl treatment for 24, 48, and 72 h, but was significantly downregulated at higher NaCl levels (100 mM, 150 mM, 200 mM) (Figure 9). At 50 mM NaCl, the relative expression of *NHX1* and *SOS1* was significantly upregulated compared to the control, whereas it was significantly downregulated at higher NaCl concentrations (100 mM, 150 mM, and 200 mM) (Figure 9).

## 4. Discussion

Salt stress is an important environmental factor that restricts plant growth and development and reduces crop yield [49]. Under salt stress, plants exhibit slow growth and development, metabolic inhibition, and, in severe cases, wilting and even death. Here, we showed that NaCl treatment significantly inhibited the growth of *R. pseudoacacia* seedlings (Figure 1). Plant height and root length decreased, dry and fresh weight also decreased with the increase of salt concentration.

Salt stress has a broad range of effects on plant metabolism, including the contents of other ions and organic soluble substances, various aspects of cell metabolism, and the expression of corresponding genes [50]. For example, in this experiment, with increased NaCl concentrations, Na^+^ and Cl^−^ accumulated in large quantities in plants, while K^+^ decreased, resulting in a high Na^+^/K^+^ ratio (Figure 2A–D). K^+^ is a key ion to ensure the normal metabolism of plants. Because of Na^+^, K^+^ competing K^+^/Na^+^ transporters are at the same binding site and the accumulation of too much Na^+^ inhibits K^+^/Na^+^ exchange, thus significantly reducing the K^+^ content in the seedlings [51]. Previous studies have shown that under NaCl stress, and more Na^+^ was retained in the roots, the K^+^/Na^+^ ratio in the above-ground part of *R. pseudoacacia* was lower, while less Na^+^ is distributed in the leaves, thus reducing the damage of Na^+^ to the leaves [52]. Under salt stress, plants tend to upregulate Na^+^/H^+^ antiporter genes to maintain Na^+^/K^+^ homeostasis and avoid Na^+^ accumulation in the cytoplasm. *SOS1*, one Na^+^/H^+^ antiporter located on the plasma membrane, can be activated by phosphorylation to transport Na^+^ to the extracellular matrix. Similarly, *NHX1*, located on the tonoplast, can be activated by phosphorylation to pump excess intracellular Na^+^ into the vacuole [53,54,55]. An increasing number of experiments have demonstrated the role of Na^+^/H^+^ antiporters in salt resistance: for example, heterologous expression of *AoNHX1* (from *Avicennia officinalis* L.) increases salt tolerance in rice and Arabidopsis [56], *RtNHX1* (from *Reaumuria trigyna*) enhances salt tolerance in transgenic Arabidopsis plants by sequestering Na^+^ into the vacuole and decreasing the Na^+^/K^+^ ratio in the cytoplasm [57]. In this experiment, the relative expression of *RpNHXl* and *RpSOS1* was higher than in the control under 50 mM NaCl treatment, but lower at higher NaCl concentrations (100 mM, 150 mM, and 200 mM) (Figure 9). This indicates that *R. pseudoacacia* seedlings under 50 mM NaCl treatment can effectively maintain ion homeostasis in the cytoplasm by either compartmentalizing Na^+^ into the vacuole or transporting Na^+^ to the extracellular matrix, thereby reducing damage caused by salt stress, but this mechanism becomes insufficient at higher salt concentrations. This is consistent with past findings [56,57].

Salt ions in soil can cause osmotic stress in plants, which prompts them to accumulate various organic substances in their roots to reduce osmotic potential, improve cellular water retention capacity, and mitigate the damage caused by osmotic stress. These substances are either small molecules such as proline and betaine, or structural substances such as sucrose and starch [51]. Organic osmoregulatory substances play an important role in plant tolerance to salt stress. Under salt stress, the *P5CS1* (*Delta1-pyrroline-5-carboxylate synthase 1*) gene in the Arabidopsis *myc2* mutant is upregulated to synthesize proline, thereby enhancing salt tolerance [58]. The enrichment of soluble sugars in rice can also effectively relieve osmoregulatory effects and enhance salt tolerance [59,60]. In this experiment, concentrations of organic substances increased under salt stress in *R. pseudoacacia* seedlings (Figure 3). The most significant increase was in the proline content, suggesting that proline plays a major role in osmotic adjustment [58,59].

Salt stress is a complex process, and almost all physiological and biochemical pathways in plants will be affected [8,61]. It not only causes direct primary damage to plants, but also causes secondary damage, such as peroxide stress [62]. In *R. pseudoacacia* seedlings subjected to salt stress, the O_2_^−^, H_2_O_2_, and MDA contents increased, as did electrical conductivity (Figure 4), which are signs of oxidative stress [63,64]. However, salt-tolerant plants tend to adapt to salt stress by increasing ROS scavenging capacity using enzymatic antioxidants (SOD, POD, APX, CAT, etc.) and non-enzymatic antioxidants (AsA, GSH, etc.) [65]. Among them, SOD is the key enzyme for O_2_^−^ scavenging, and CAT, APX, and POD are the key enzymes for H_2_O_2_ scavenging [10,66]. Under salt stress, cucumber responded by adjusting CAT and APX antioxidant enzyme activities [67], whereas the CAT and SOD activities in sweet sorghum increased and then decreased under NaCl stress [68]. In our experiments, the activities of SOD, POD, CAT, and APX increased at low concentrations of NaCl, but decreased at high concentrations of NaCl (Figure 5). This indicates that *R. pseudoacacia* seedlings can tolerate low concentrations of NaCl and effectively scavenge ROS by increasing antioxidant enzyme activity. However, at high concentrations of NaCl, the ROS scavenging ability of seedlings decreased, resulting in ROS accumulation and subsequent damage to the plants’ growth and development [67,68,69].

Photosynthesis is sensitive to salt stress [16]. Pn, Ci, and Gs are important for understanding physiological processes of leaves in nature; changes in photosynthetic parameters are direct reflections of photosynthetic function [70]. In this experiment, Pn, Tr, and Gs of *R. pseudoacacia* seedlings decreased under NaCl treatment, indicating that photosynthesis was inhibited (Figure 6). When plants are exposed to salt stress, the first reaction is stomatal closure caused by osmotic stress, the decrease of stomatal conductance Gs causes the decrease of stomatal factor and Ci, thus limiting Pn [71]. Furthermore, chlorophyll content is an important indicator of leaf senescence. Our study showed that chlorophyll content was significantly reduced under NaCl treatment (Figure 7), which likely contributed to reduced photosynthesis. Under salt stress, there is a large amount of Na^+^ uptake, chlorophyll loss, and stomatal closure in *R. pseudoacacia* seedlings. The resulting restriction of CO_2_ entry and salt accumulation led to the inhibition of CO_2_ assimilation. Because chloroplasts serve as the main site of photosynthesis, the structural and functional integrity of chloroplasts is another prerequisite for photosynthesis [72]. Chloroplasts are also the organelles most sensitive to salt stress, which often damages the chloroplast membrane system and deforms the chloroplast structure. In *Cornus hongkongensis* subsp. *elegans*, *Vitis amurensis* Rupr, and *Podocarpus macrophyllus*, salt stress led to disorganized cystoid, basidiome, and stroma lamellae structures, which seriously damaged the integrity of the chloroplast ultrastructure [17,27,73]. The results of our experiments showed that the grease content increased as the salt concentration increased. Additionally, the chloroplast ultrastructure in *R. pseudoacacia* seedlings was clearly damaged by salt stress, with the chloroplasts being distorted and deformed, and the starch granules swollen and irregular (Figure 8). Previous reports showed that *Atriplex halimus* may form lipid deposition to resist the harmful effects of salt-induced toxicity [74,75]. The chloroplast ultrastructure was consistent with previous reports of salt-treated diploid black locust [71].

The photosynthetic complex PSII is an important site for the absorption, transmission, and conversion of light energy in the light reactions, and the core proteins D1, V2, and D2 are encoded by the chloroplast genes *psbA*, *psbC*, and *psbD*. In this experiment, the expression of *psbA*, *psbC*, and *psbD* was significantly upregulated under 50 mM NaCl treatment (Figure 9), indicating that *R. pseudoacacia* seedlings tolerated low concentration NaCl stress. However, these genes were downregulated at elevated salt concentrations, suggesting that these core PSII proteins are important factors affecting photosynthetic efficiency during salt stress. *psaA* has a light-induced regulatory function during the transition from proplastid to chloroplast in C4 plants (such as sorghum) [76]. In the present study, the expression of *psaA*, *psaB*, and *psaC* was also significantly upregulated under low-concentration NaCl treatment but was significantly downregulated under high-concentration NaCl treatment (Figure 9). We speculate that the upregulation of these genes at lower NaCl concentrations can compensate for the damage to chloroplasts caused by salt stress, but this compensatory mechanism becomes insufficient at higher NaCl concentrations.

We also found that the expression of *rps7*, which encodes a chloroplast ribosomal protein that plays an important role in maintaining chloroplast function, was significantly upregulated under low-concentration NaCl treatment but significantly downregulated under high-concentration NaCl treatment (Figure 9), indicating that excess NaCl severely affected the synthesis of chloroplast ribosomal proteins. The NADH dehydrogenase-like (NDH) complex is involved in photosynthetic electron transport chains and catalyzes the transfer of electrons to drive the production of adenosine triphosphate [77]. It is encoded by the chloroplast genome and contributes to the adaptation of chloroplasts to environmental stresses and plays an important role in photosynthetic efficiency and response to stress [28]. The expression of *ndhE* and *ndhH*, encoding an NADH subunit and the PSⅠ psaC subunit, respectively [78], was significantly downregulated under high concentrations of NaCl (Figure 9). We presume that these high NaCl concentrations severely disrupted signaling processes regulated by *ndhE* and *ndhH*, preventing the synthesis of the NDH complex. Several studies have demonstrated a strong relationship between *ropA* expression and abiotic stresses. In Arabidopsis, low levels of oxidative stress activate *ropA*, leading to the production of H_2_O_2_ and ethanol dehydrogenase to resist hypoxic stress [79,80]. In the present experiment, the expression of *ropA* was upregulated at low concentrations of NaCl, indicating that *R. pseudoacacia* seedlings are tolerant of low-concentration NaCl treatment (Figure 9), which is consistent with the antioxidant results. However, at high concentrations of NaCl (100 mM, 150 mM, and 200 mM), the above genes regulating chloroplast development were significantly downregulated, affecting the development and structure of chloroplasts and leading to structural abnormalities and functional damage.

## 5. Conclusions

In all, our results showed that *R. pseudoacacia* seedlings can tolerate low levels of NaCl. Although biomass decreased, photosynthetic structures were not damaged, and the plants continued to grow in the presence of low concentrations of NaCl. However, the high-concentration NaCl treatment downregulated ion transport- and chloroplast development-related gene expression, leading to ion accumulation and damage to photosynthetic structures, and thus resulting in growth arrest and sometimes death. These results will provide theoretical guidance for planting *R. pseudoacacia* on saline-alkali land.

## Figures and Tables

**Figure 1 plants-12-01283-f001:**
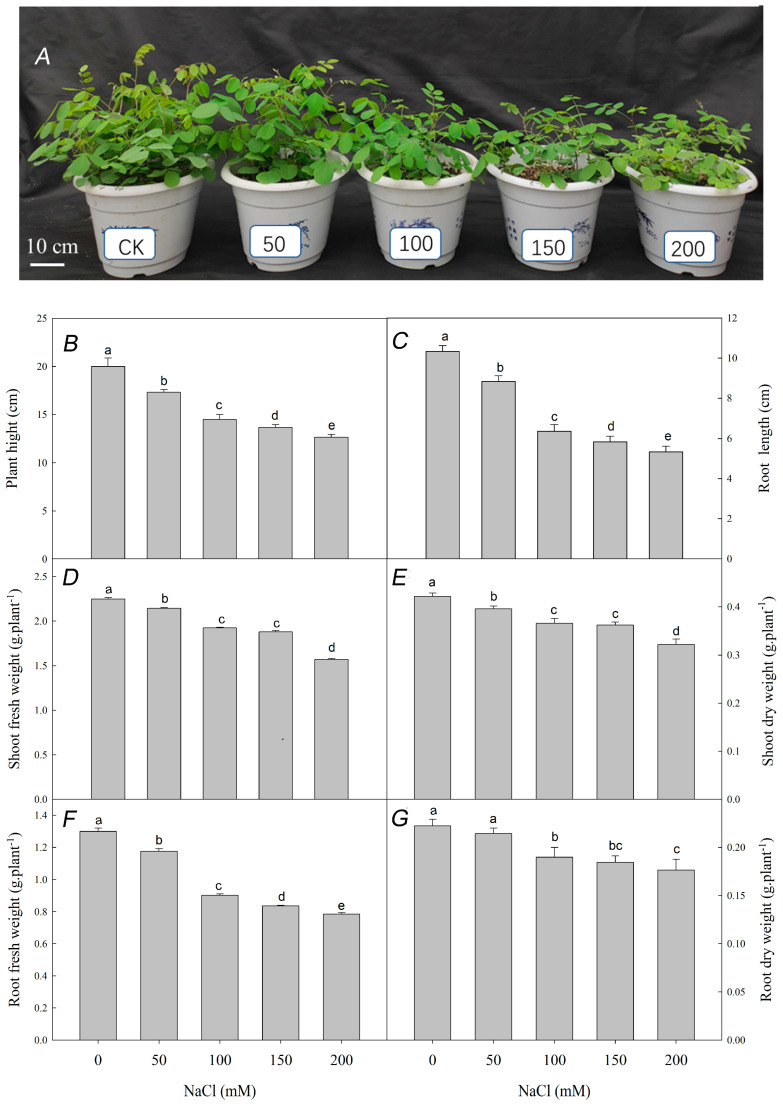
Effect of NaCl stress on the growth of *R. pseudoacacia* seedlings. (**A**) Seedlings under different NaCl concentrations (0 mM, 50 mM, 100 mM, 150 mM, 200 mM); (**B**) plant height; (**C**) root length; (**D**) shoot fresh weight; (**E**) shoot dry weight; (**F**) root fresh weight; (**G**) root dry weight. Values are mean ± SD of five biological replicates. Bars with different letters are significantly different at *p* < 0.05 according to Duncan’s multiple range tests.

**Figure 2 plants-12-01283-f002:**
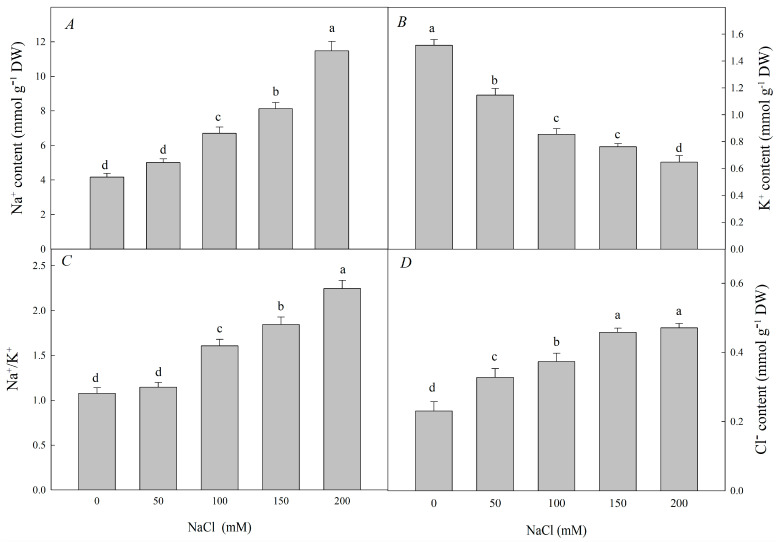
Effect of NaCl stress on Na^+^, K^+^, Cl^−^, and Na^+^/K^+^ ratio and of *R. pseudoacacia* seedlings. (**A**) Na^+^ content; (**B**) K^+^ content; (**C**) Na^+^/K^+^; (**D**) Cl^−^ content. Values are mean ± SD of five biological replicates. Bars with different letters are significantly different at *p* < 0.05 according to Duncan’s multiple range tests.

**Figure 3 plants-12-01283-f003:**
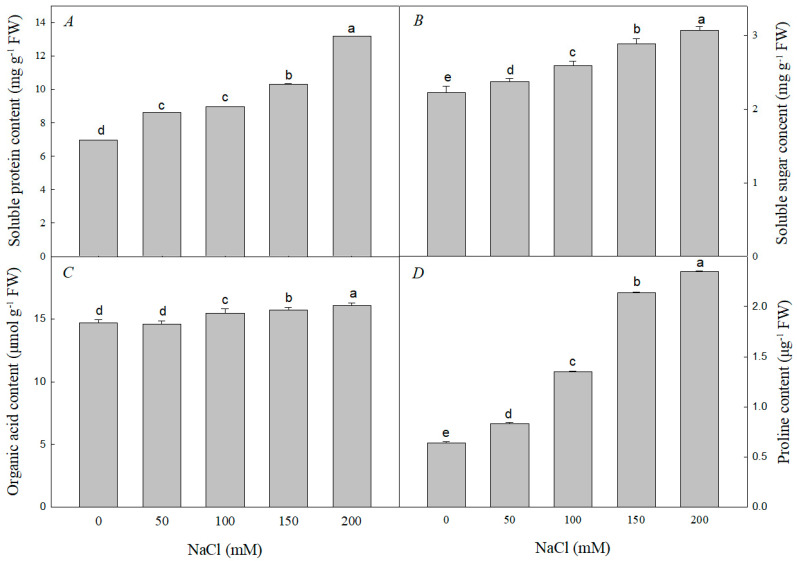
Effect of NaCl stress on the contents of soluble protein (**A**), soluble sugar (**B**), organic acids (**C**), and proline (**D**) of *R. pseudoacacia* seedlings. Values are mean ± SD of five biological replicates. Bars with different letters are significantly different at *p* < 0.05 according to Duncan’s multiple range tests.

**Figure 4 plants-12-01283-f004:**
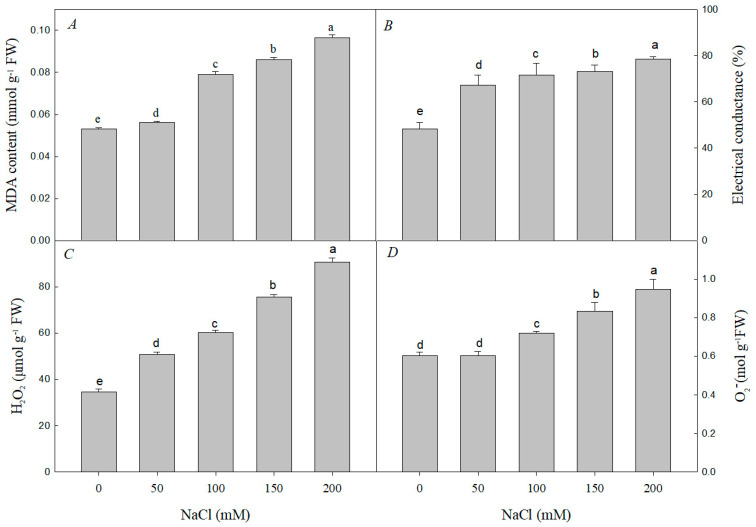
Effect of NaCl stress on the MDA content (**A**), electrical conductance (**B**), H_2_O_2_ content (**C**), and O_2_^−^ content (**D**) of *R. pseudoacacia* seedlings. Values are mean ± SD of five biological replicates. Bars with different letters are significantly different at *p* < 0.05 according to Duncan’s multiple range tests.

**Figure 5 plants-12-01283-f005:**
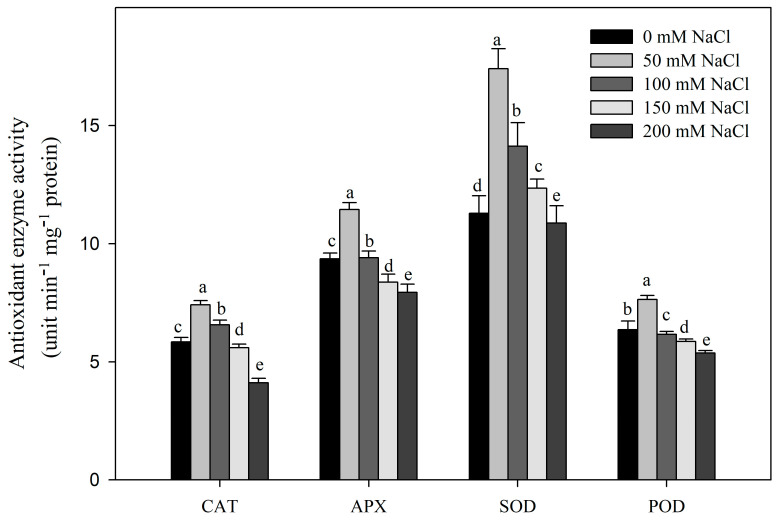
Effect of NaCl stress on activities of CAT, SOD, APX, and POD in *R. pseudoacacia* seedlings. Values are mean ± SD of five biological replicates. Bars with different letters are significantly different at *p* < 0.05 according to Duncan’s multiple range tests.

**Figure 6 plants-12-01283-f006:**
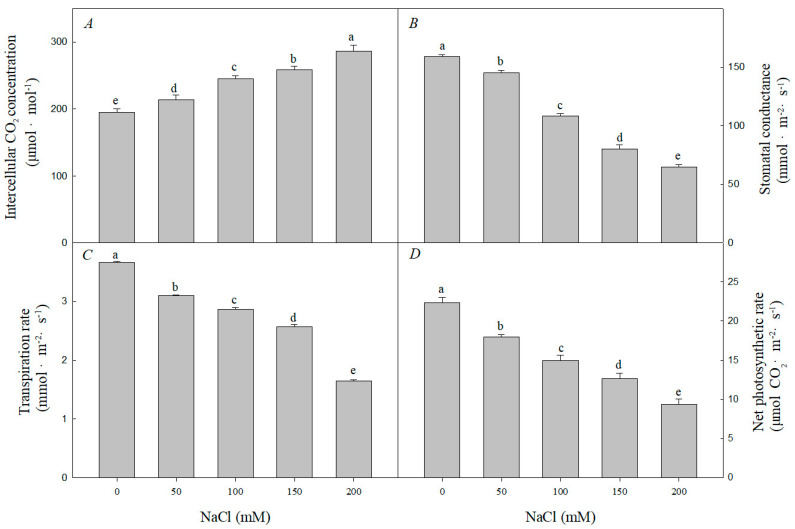
Effect of NaCl stress on photosynthesis of *R. pseudoacacia* seedlings. (**A**) Intercellular CO_2_ concentration; (**B**) stomatal conductance; (**C**) transpiration rate; (**D**) net photosynthetic rate. Values are mean ± SD of five biological replicates. Bars with different letters are significantly different at *p* < 0.05 according to Duncan’s multiple range tests.

**Figure 7 plants-12-01283-f007:**
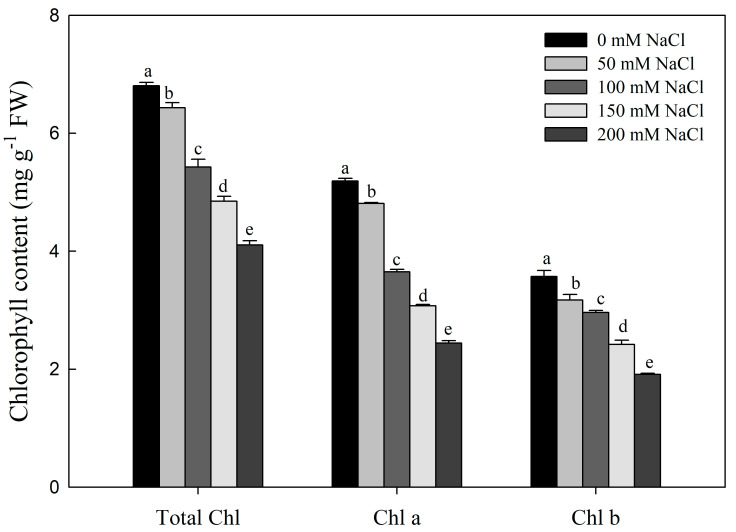
Effect of NaCl stress on the content of total chlorophyll, chlorophyll a, and chlorophyll b in *R. pseudoacacia* seedlings. Values are mean ± SD of five biological replicates. Bars with different letters are significantly different at *p* < 0.05 according to Duncan’s multiple range tests.

**Figure 8 plants-12-01283-f008:**
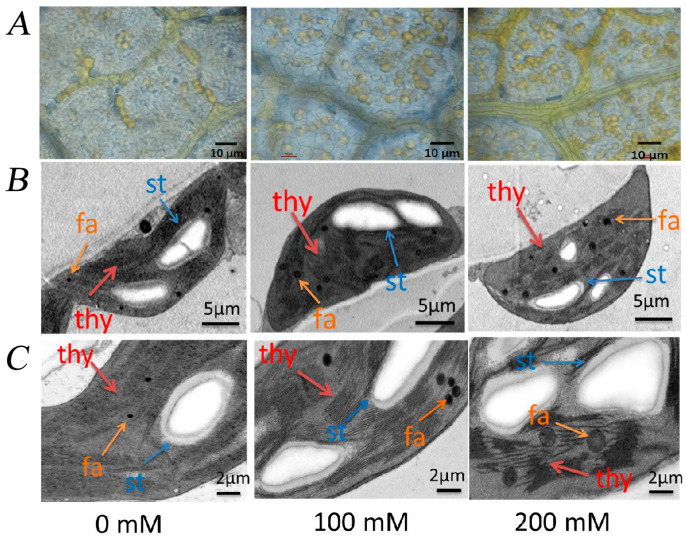
Effect of NaCl stress on the grease content (**A**) and chloroplast structure (**B**,**C**) of *R. pseudoacacia* seedlings. thy: thylakoid; st: starch granule; fa: liposphere. Bars in (**A**) are 10 µm; bars in (**B**) are 5 µm; bars in (**C**) are 2 µm.

**Figure 9 plants-12-01283-f009:**
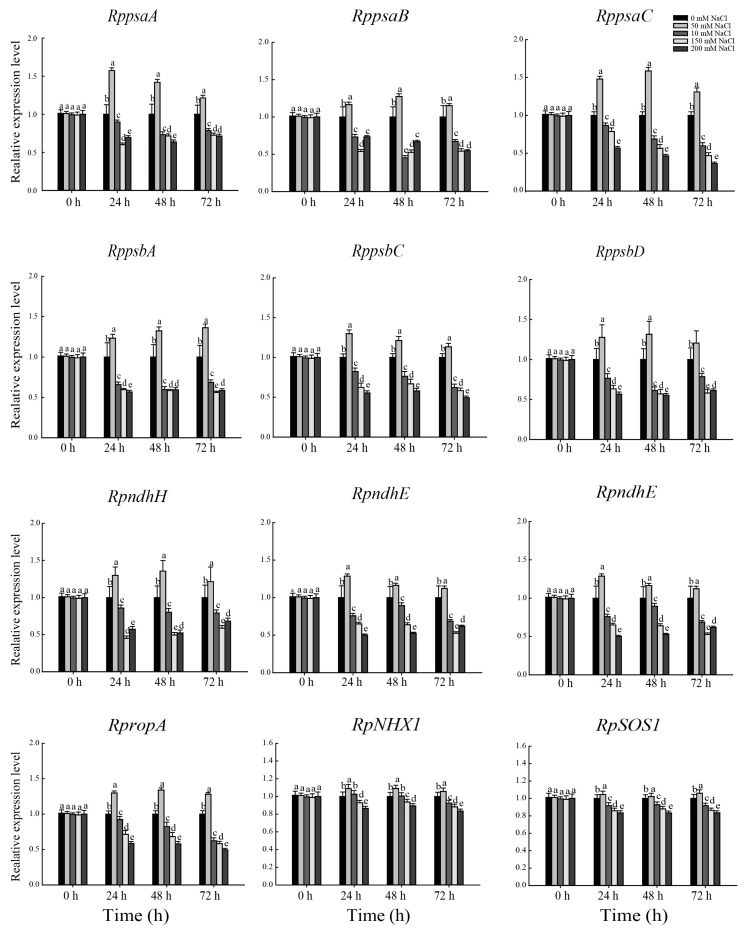
Effect of NaCl stress on the expression of key genes in chloroplast development (*RppsaA*, *RppsaB*, *RppsaC*, *RppsbA*, *RppsbC*, *RppsbD*, *RpndhE*, *RpndhH*, *Rprps7*, and *RpropA*) and ion transporters (*RpNHX1* and *RpSOS1*) of *R. pseudoacacia* seedlings. Values are mean ± SD of five biological replicates. Bars with different letters are significantly different at *p* < 0.05 according to Duncan’s multiple range tests.

## Data Availability

The data are contained within the article.

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
