# Peer review of "Salt Stress Inhibits Photosynthesis and Destroys Chloroplast Structure by Downregulating Chloroplast Development–Related Genes in Robinia pseudoacacia Seedlings"

_plants, 2023, doi:10.3390/plants12061283_

Round 1

Reviewer 1 Report

Dear Editor

Thank you very much for your invitation to review this manuscript:

Salt stress inhibits photosynthesis and destroys chloroplast structure by downregulating chloroplast development–related genes in Robinia pseudoacacia seedlings

The work is reported on the negative effect of salt stress on photosynthesis and chloroplast structure as well as chloroplast development–related genes in Robinia pseudoacacia seedlings.

I read the wark carefully, it introduces very important information.

The authors should follow the comments in pdf version and in the following lines:

·         Lines 40, 42, 44 The authors must follow the journal rules (in the introduction section delete the author names and add No instead)

·         Line 46, 48 (delete the author names and add No instead).

·         Revise the introduction section (delete the author names and add No instead)

·         Line 146, revise, it is not Li et al. method

·         Line 154 delete malondialdehyde and add reference

·         Line 161, reference needed

·         Lines 169,175,182,188, references needed

·         Line 175, add the details about SOD determination and other enzymes

·         Line 182, write the wavelengths

·         Line 217, reference needed

·         Line 308 delete

·         Line 456 revise all the references and follow the rules

·         Please see the pdf version

Best regards

Author Response

Point 1    Lines 40, 42, 44,46,48  The authors must follow the journal rules (in the introduction section delete the author names and add No instead)

Response: Thanks for the suggestions, we have accepted your advice and re-edited the references and replaced the author's name with No.. 

Point 2    Line 46, 48 (delete the author names and add No instead).

Response:  Done.

Point 3       Revise the introduction section (delete the author names and add No instead)

Response:  Done.

Point 4       Line 146, revise, it is not Li et al. Method

Response: Thanks for the suggestions. We have accepted your advice and add the reference in line 146,151,157, marked in red in the manuscript.

Point 5         Line 154 delete malondialdehyde and add reference

Response:  Done in line 150 and marked in red in the manuscript.

Point 6·         Line 161, reference needed

Response:  Done in line 157 and marked in red in the manuscript.

Point 7·         Lines 169,175,182,188, references needed

Response:  Done in line 173, 176,180, 183 and marked in red in the manuscript.

Point 8·         Line 175, add the details about SOD determination and other enzymes

Response:  Thanks for the suggestions. We have accepted your advice and add the details about SOD and POD determination in line 173-178. Marked in red in the manuscript.

Point 9·         Line 182, write the wavelengths

Response:  Done in line 183 and marked in red in the manuscript.

Point 10·         Line 217, reference needed

Response:  Done in line 222 and marked in red in the manuscript.

Point 11·         Line 308 delete

Response:  Done

Point 12·         Line 456 revise all the references and follow the rules

Response:  Thanks for the suggestions. We have accepted your advice and re-edited the references.

Reviewer 2 Report

Dear authors,

In this paper the authors describe how different photosynthetic parameters, the structure of the chloroplast, and different genes in R. pseudoacacia are affected by high concentrations of NaCl. I consider that the work is well done, with a lot of experimental work, the data obtained are of interest because not too much research on saline stress has been carried out on this plant. However, there are a number of errors, some of which are important and must be corrected by the authors.

The introduction is very well planned and written.

L82: Huang et al. Year??

L146: “ Li et al. (2022) (Li et al., 2022)” Check for this type of error in all text

L107: Li et al. 2020 it is repeated

L113: “Measurements were taken after 2 weeks at  that concentration.” In the abstract, line 20, it says that the measurements were made at 4 weeks? clarify please.

L171: “. Antioxidant Activity Aassays” Typo

L175: Why SUPEROXIDE DISMUTASE and the other enzymes in the paragraph are named in capital letters?

L175-L182: Indicate the correct reference of such methods

L132:” pH 7.8)” Typo

L135: “(Lin et al., 2018)).” Typo

The caption of Fig 1 is wrong, it seems that photo A was inserted and the letters A-F were no longer changed. In the photo of the pots indicate that they are concentrations of NaCl. Put the day on which the photo was taken and the day on which the rest of the parameters were studied, please.

The title of the section "3.1. Effect of Nacl Stress on the Growth, Ions Content and Organic Soluble Substances Contents  of R. Pseudoacacia Seedlings" does not say that it is going to talk about the expression of the genes, but the HhNHX1 and genes are analyzed by RT-PCR HhSOS1 (Fig 2 -F). This is confusing. It is not explained why it is done this way. Explain why, or change the title of the section, or better yet, put the RT-PCR data of these genes in another section where only gene expression results are discussed.

L262:” H2O2 content” typo

L263: “O2 content” typo

The caption of figure 8 is completely wrong. The photos are not numbered, nor is it indicated what each of the abbreviations is, there is no data on the grease content anywhere.

The caption of figure 9. I do not understand how the collection of samples in Figure 9 and 2 for the RT-PCR could be carried out, it is not indicated in the materials and methods. When were the samples taken? What is zero time? How was the sample taken for 24, 48 and 72 hours, were these samples taken from the same plant?

* The name of all genes must always appear in italics.

* It is convenient to include a conclusions section, which could be done from the last paragraph of the conclusions

L331:” genes(Liu et al., 2018” Typo

L337: “(NHX1) ” Typo

L390:” Pn, Tr, and Gs” These abbreviations have not been defined before

L397: “Podocarpus macrophyllus.” Typo

L410: PSI? proteins

L423:” NADH complex” what is this? define better please

L425:” plays an important role in photosynthetic efficiency and response to stress” Reference needed

* I see the discussion well raised and carried out, however, this plant has already been used previously for some studies of saline stress, I consider that such works would enrich the discussion, I quote two: doi: 10.1038/srep23098. https://pubmed.ncbi.nlm.nih.gov/21812288/

Author Response

Response to Reviewer 2 Comments

Point1  L82: Huang et al. Year??

Response: Thanks for the suggestions, we have accepted your advice and add the year in line 79.

Point2  L146: “ Li et al. (2022) (Li et al., 2022)” Check for this type of error in all text

Response: Thanks for the suggestions, we have accepted your advice and check for this type of error in all text. Marked in red in line 103-105,141.

Point 3  L107: Li et al. 2020 it is repeated

Response: Done in line 103 and marked in red in the manuscript.

Point 4  L113: “Measurements were taken after 2 weeks at  that concentration.” In the abstract, line 20, it says that the measurements were made at 4 weeks? clarify please.

Response: Thanks for the suggestions, we have checked this error and correct. Marked in red in the manuscript.

Point 5   L171: “. Antioxidant Activity Aassays” Typo

Response: Done

Point 6  L175: Why SUPEROXIDE DISMUTASE and the other enzymes in the paragraph are named in capital letters?

Response: Thanks for the suggestions, we have checked this error and correct. Marked in red in line 172,173,175, 180 in the manuscript.

Point 7   L175-L182: Indicate the correct reference of such methods

Response: Thanks for the suggestions, we have added related references in line175-182 in the manuscript.

Point 8   L132:” pH 7.8)” Typo

Response: Done

Point 9   L135: “(Lin et al., 2018)).” Typo

Response: Done

Point 10  The caption of Fig 1 is wrong, it seems that photo A was inserted and the letters A-F were no longer changed. In the photo of the pots indicate that they are concentrations of NaCl. Put the day on which the photo was taken and the day on which the rest of the parameters were studied, please.

Response: Thanks for the suggestions, we have re-edited the figures in the article and increased the figures resolution. The photo was taken on 2021.05.12, plant height, root length, and fresh weight were measured on the same day. Dry weight was measured on 2021.05.20.

Point 11  The title of the section "3.1. Effect of Nacl Stress on the Growth, Ions Content and Organic Soluble Substances Contents  of R. Pseudoacacia Seedlings" does not say that it is going to talk about the expression of the genes, but the HhNHX1 and genes are analyzed by RT-PCR HhSOS1 (Fig 2 -F). This is confusing. It is not explained why it is done this way. Explain why, or change the title of the section, or better yet, put the RT-PCR data of these genes in another section where only gene expression results are discussed.

Response: Thanks for the suggestions. We have accepted your advice and put the RT-PCR data of NHX1 and SOS1 genes in the section 3.4, shown in Figure 9.

Point 12  L262:” H2O2 content” typo

Response: Done and marked in red in line 264 in the manuscript. 

Point 13  L263: “O2 content” typo

Response: Done and marked in red in line 264 in the manuscript.

Point 14 The caption of figure 8 is completely wrong. The photos are not numbered, nor is it indicated what each of the abbreviations is, there is no data on the grease content anywhere.

Response: Thanks for the suggestions. We have re-edited the figures in the article , increased the figures resolution and add some description about grease content in line 306-310 in the manuscript. 

Point 15  The caption of figure 9. I do not understand how the collection of samples in Figure 9 and 2 for the RT-PCR could be carried out, it is not indicated in the materials and methods. When were the samples taken? What is zero time? How was the sample taken for 24, 48 and 72 hours, were these samples taken from the same plant?

Response: Thanks, we have described the sampling procedure in detail in line 205-208 and marked in red in the manuscript. 

Point 16  * The name of all genes must always appear in italics.

Response: Done, we have checked all the manuscript and corrected them in line 360, 362.

Point 17 * It is convenient to include a conclusions section, which could be done from the last paragraph of the conclusions

Response: Thanks for the suggestions. We have accepted your advice and add conclusions section in line 468.

Point 18  L331:” genes(Liu et al., 2018” Typo

Response: Done

Point 19  L337: “(NHX1) ” Typo

Response: Done

Point 20  L390:” Pn, Tr, and Gs” These abbreviations have not been defined before

Response: Thanks for the suggestions, We defined these abbreviations in line 284-285 and marked in red in the manuscript. 

Point 21  L397: “Podocarpus macrophyllus.” Typo

Response: Done

Point 22  L410: PSI? Proteins

Response: Done and marked in red in the manuscript in line 436.

Point 23  L423:” NADH complex” what is this? define better please

Response: Thanks for the suggestions,We agree with you and described in detail in 449-451. Marked in red in the manuscript.

Point 24  L425:” plays an important role in photosynthetic efficiency and response to stress” Reference needed

Response: Done in line 454.

Point 25  * I see the discussion well raised and carried out, however, this plant has already been used previously for some studies of saline stress, I consider that such works would enrich the discussion, I quote two: doi: 10.1038/srep23098. https://pubmed.ncbi.nlm.nih.gov/21812288/

Response: Thanks for the suggestions. We read and consulted the two study and recent reference. The details are provided in line 344-350,401-408,419-428. Marked in red in the manuscript.

Round 2

Reviewer 2 Report

I believe that the authors have responded favorably to most of my suggestions and I accept the paper in its current version.